# Just Breathe: Improving LEP Outcomes through Long Interval Breathing

**Andrew Wold** [1,*], **Rebecca Boehme** [2] and **Magnus Thordstein** [3,4]

1. Faculty of Medicine and Health, Örebro University, 70182 Örebro, Sweden
2. Center for Social and Affective Neuroscience, Department of Biomedical and Clinical Sciences, Linköping University, 58185 Linköping, Sweden; rebecca.bohme@liu.se
3. Neuro, Department of Biomedical and Clinical Sciences, Linköping University, 58185 Linköping, Sweden; magnus.thordstein@liu.se
4. Department of Clinical Neurophysiology, University Hospital, 58185 Linköping, Sweden
* Correspondence: andrew.wold@liu.se

**Abstract:** Background: Laser-evoked potentials (LEPs) constitute an objective clinical diagnostic method used to investigate the functioning of the nociceptor system, including signaling in thin peripheral nerve fibers: Aδ and C fibers. There is preliminary evidence that phase locking LEPs with the breathing cycle can improve the parameters used to evaluate LEPs. Methods: We tested a simple breathing protocol as a low-cost improvement to LEP testing of the hands. Twenty healthy participants all underwent three variants of LEP protocols: following a video-guided twelve-second breathing instruction, watching a nature video, or using the classic LEP method of focusing on the hand being stimulated. Results: The breath protocol produced significantly shorter latencies as compared with the nature or classic protocol. It was also the least prone to artifacts and was deemed most acceptable by the subjects. There was no difference between the protocols regarding LEP amplitudes. Conclusions: Using a breathing video can be a simple, low-cost improvement for LEP testing in research and clinical diagnostics.

**Keywords:** LEP; neurophysiology; breathing; evoked potentials; pain

## 1. Introduction

Laser-evoked potentials (LEPs) play a critical role in clinical practice as well as in research routines. LEPs are used to investigate the pathophysiological mechanisms underlying various types of chronic pain [1], and are considered the most accurate diagnostic test for selectively assessing the nociceptive system [2], especially for investigating patients with neuropathic pain conditions, such as small fiber neuropathy [3]. It works by applying laser stimulation that selectively activates small fiber, Aδ and C mechano-heat, within the superficial layers of the skin. This stimulation, in combination with recordings from scalp electrodes, can generate reproducible evoked potentials. In neurophysiological examinations, LEPs can be used to assess signal conductance, thereby contributing to the detection of small nerve fiber dysfunctions, for example, in chronic pain, headaches and even in psychiatric conditions [4]. In research, LEPs have been used for studies on pain perception [5,6] and on the neurobiological processing of painful stimuli [7].

In the clinical setting, LEPs plays an important role, since somatosensory processing cannot be fully characterized using somatosensory evoked potentials only. Somatosensory evoked potentials are measured in response to the stimulation of large fibers, signal transfer through the dorsal column, synapsing in the brain stem and thalamus, thalamocortical projection [8] and cortical activation. Small nerve fiber signals are processed differently: their first relay station is in the spinal cord, not in the brainstem, from which the information is conveyed via the contralateral spinothalamic tract to the thalamus [9]. Additionally, in the brain, pain stimuli can be processed differently, e.g., secondary sensory cortex and

insula, compared to non-painful somatosensory stimuli [10]. Taken together, LEPs play a central role in the accurate diagnosis and clinical investigations of small fiber function.

In clinical practice, LEP responses are elicited via the stimulation of the skin with short pulses of an infrared laser light. LEP stimulation strength is adjusted to individual thresholds and causes a short, pinprick-like pain. Typically, a subject receives a train of stimulations, the LEP responses are averaged and, for diagnostic purposes, these averages are compared to standard values to determine if transmission dysfunction is present. This protocol implies that LEP responses are reasonably reliable and invariable. Doubt on this assumption was cast by studies indicating that LEP amplitudes were influenced by attention [11] to and predictability [12] of the stimulation, as well as vigilance of the subjects [13]. In order to adjust clinical protocols to these findings, the stimulator is moved slightly for every pulse, thereby stimulating slightly different locations in the same dermatome, and subjects are asked to look at the stimulator in order to keep their attention on the pain stimulation [14].

Bringing about a new challenge, studies have found that the detection, perception and processing of various exteroceptive stimuli depend on basic physiological processes in the body: heart rate [15] and breathing phase. For example, near-threshold tactile stimuli are detected more easily if they occur at the end of the cardiac cycle [16], which is accompanied by higher somatosensory evoked potential amplitudes [17]. Similar effects have been reported for other sensory modalities, e.g., vision [18]. Respiration and cardiac cycles are closely linked. Additionally, for respiration, an interaction of the cycle phase and perception [19], as well as cognitive processing, have been demonstrated [20,21]. Depriving breathing in an air hunger LEP study showed an effect on LEP amplitudes, providing a mechanism through which air satiation could possibly improve LEP outcomes [22]. One study demonstrated, using intraepidermal electrical stimulation, an interaction of the perceived pain intensity and the evoked cortical potential amplitudes, but not latencies, with the breathing phase [23].

Based on these previous findings, we hypothesized that LEP responses in healthy adults would be improved when attention is fixed on a simple task and would synergistically interact when subjects engage in a calming, slow rhythm breathing exercise. In other words, following a breathing instructional video should improve the two main outcome variables, amplitude and latency, compared to the classic LEP protocol. We, therefore, set out to compare a classical clinical protocol of LEPs with a variant of this protocol, where subjects were instructed to breath at a certain rate. As a control condition for the potential effects of the calming video, which led the breathing exercise, subjects watched a nature video without breathing guidance. Such a breathing protocol has the potential to improve the consistency and, therefore, the diagnostic capability of LEPs, without complicating the procedure, and to reduce the discomfort of the examinee.

## 2. Methods

### 2.1. Procedure

The study received ethical approval (Dnr 2016/433-31) and was conducted in accordance with the declaration of Helsinki. Twenty participants (7 female; median age = 35; age range: 22–60) volunteered to undergo three variations of a LEP protocol. The sample size was determined by using G*Power software (version 3.1.9.7, Dusseldorf, Germany) with parameters for a matched pair *t*-test design using a one-tailed alpha value of 0.05 and a power level of 0.85, with an effect size of 0.635, which was derived from the mean reported statistically significant effect sizes of a LEP breathing restriction study from Dangers et al. (2015) [22]. The inclusion criteria were a self-reported full health nerve status and toleration of the lowest LEP setting. Exclusion criteria were previous nerve damage in either the hand, arm or shoulder, or any neuropathy symptoms in the upper extremities (tingling,

numbness, or temperature sensitivities). The participants determined a tolerable LEP intensity and had scalp electrodes attached to them to record evoked potentials (see Section 2.2). This was a within-subject design with a pseudo-randomized order of the three protocols: breathing, nature, or classic (see Section 2.3). A starting hand was randomly assigned, after which the hand being stimulated was always alternated until each of the three stimulation protocols was performed on each hand (six in total). To evaluate adherence to the breathing protocol, the breathing rate was measured per twelve-second stimulus cycle. After each session, participants were asked to use a five-point scale to assess the ease of the condition (1—very difficult; 5—very easy), and the agreeableness of the experience (1—very disagreeable/uncomfortable; 5—agreeable/neutral experience). For agreeableness, we did not expect LEPs to be rated as a positive experience and, therefore, decided to set the high end of the scale as neutral. Finally, after all six sessions were completed, participants were asked which protocol they most preferred. The whole procedure took approximately one hour.

### 2.2. Laser Evoked Potentials

We used a neodymium: yttrium–aluminum–perovskite (Nd; YAP) laser (Stimul 1340, DEKA Ltd., Calenzano, Italy) set to 0.5 Hz, 10 ms pulse and 4 mm spot size. To determine individual suprathreshold stimulus intensity, we began stimulation at 1.5 J intensity and increased or decreased this by 0.5 J until participants reported a self-assessed pain intensity of four to five on a 10-point pain scale (where 0 indicates no felt sensation and 10 indicates extreme pain). The stimulation never exceeded 2.5 J and followed recommended guideline 1. To record potentials, we used standard software (Curry 8, Compumedics Neuroscan, Abbotsford, VIC, Australia) along with 12 electrodes connected to a 64-channel amplifier (SynAmps RT, Compumedics Neuroscan, Charlotte, NC, USA). We fixed nine scalp electrodes corresponding to Fz, Cz, Pz, T3, T4, A1, A2, a ground electrode (between Cz and T4) and an ECG detection electrode (between Pz and T3). We also placed an electrode below each eye and one above the nasion to facilitate the removal of blink artifacts. All channels had a maximum impedance of 10μV and were band pass filtered (1 to 30 Hz). In addition, we placed a dipole temperature sensor (Pro-Tech Service, Inc., Heredia, Costa Rica) under the nose to track breathing rate, which was used to evaluate adherence to the breathing protocol. After each stimulus, the point for laser stimulation was shifted to prevent fatigue and burn lesions using an interstimulation interval of twelve seconds, $(+/-2.5$ s). Each participant had both hands stimulated 20 times per protocol. All LEP stimulations were timed using a customized script (Python 3.8, Amsterdam, The Netherlands) that paired looping protocol videos with the onset of the 20 LEP stimulations.

### 2.3. Protocols

Participants were asked to undergo three separate protocols: breathing, nature and classic. For the breathing protocol, participants were instructed to watch a video loop where a circle expanded for six seconds and then shrunk for six seconds. This video was custom made for this purpose and is freely available [24]. They were informed to take all six seconds to fully breathe in, from no air to 100% capacity, and to reverse this process during a six-second exhale. For the nature protocol, participants were instructed to watch a twelve-second looping nature video of a drone moving towards trees and away from trees, similar to the motion of the expanding and contracting circles in the breathing video. No additional instructions were given. The classic protocol followed the clinical standard procedure, where participants were asked to focus on the actively stimulated hand while it was being stimulated. LEPs use a targeting laser that flutters during active stimulation; therefore, if they watched carefully, participants received a brief visual warning that the stimulus was coming.

*2.4. Analysis*

Our outcomes of interest were N2-P2 amplitude in microvolts and shortest peak N2 latency in milliseconds. Our further measures of interest were the number of breaths taken within each twelve-second stimulation cycle, the number of artifact-free cycles (of a possible 20), the rating on the ease of protocol and acceptableness and the favored protocol. To determine the amplitude and latency, we used a custom script (Python 3.8, Amsterdam, The Netherlands), which allowed the visual inspection of each stimulation and breathing count across the whole trial. The LEP recordings were performed using a standard technique aiming to define N2 and P2 as the largest negative and positive peaks in the post-stimulus interval 0–500 ms, respectively [25]. Our script has a built-in blink detection artifact filter for any peak eye channel stimulus greater than 100 µV occurring between 100 and 500 ms post stimulus. Visual inspection was used to determine if a blink artifact could have influenced a recording that was outside of automatic thresholding, for example, a blink before 100 ms that still effected the 100 to 500 ms recording window, in which case, it too was removed. Where artifacts occurred, we removed the stimulation from the total set and took the corrected average for the max N2-P2 amplitude and shortest N2 latency. No baseline correction was performed. We then averaged latency and amplitude values of the left and right sides for each protocol, and compared breathing against the control conditions of the nature and classic protocols. We expected the breathing protocol to be superior to the other two protocols, i.e., evoking larger amplitudes and shorter latencies. through higher consistency Where normality assumptions were met, we used a directed paired *t*-test (*t*-score) based on our hypothesis that the breathing protocol would perform better than other protocols. Where normality was violated or data were ordinal in structure, we used the Wilcoxon signed-rank test (z-score). All statistical analyses were performed using standard software (SPSS version 28, IBM Corp., Armonk, NY, USA).

**3. Results**

Breathing protocol amplitudes showed no statistically significant differences from either the nature ($z_{19} = -0.262$, $p = 0.397$; see Figure 1) or classic protocol ($t_{19} = 0.312$, $p = 0.378$; see Figure 1). Breathing protocol latencies were significantly shorter compared to both nature ($t_{19} = -1.826$, $p = 0.042$; see Figure 2) and classic ($t_{19} = -1.801$, $p = 0.044$; see Figure 2) protocols. The nature protocol did not show a difference in either amplitude ($z_{19} = -0.579$, $p = 0.281$) or latency ($t_{19} = -0.571$, $p = 0.287$) compared to the classic protocol.

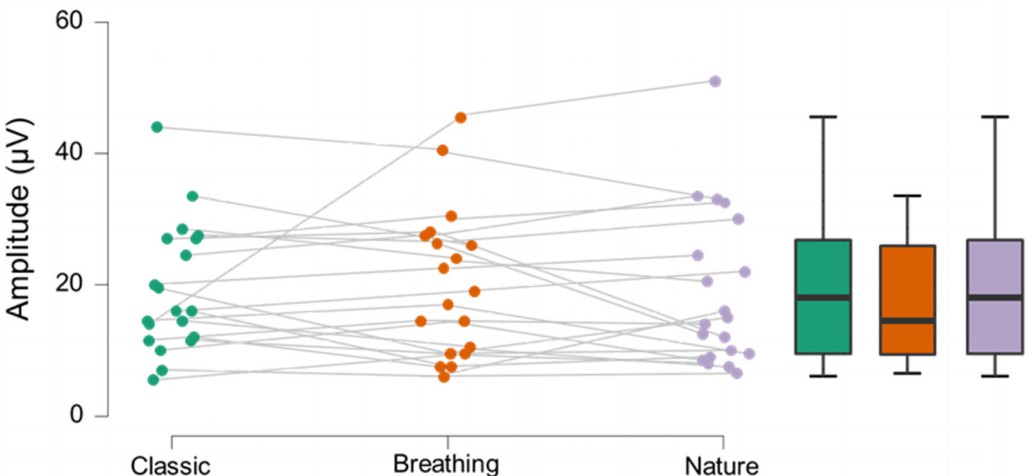

**Figure 1.** Individual and box plot data for protocol amplitudes in microvolts. No significant difference was observed.

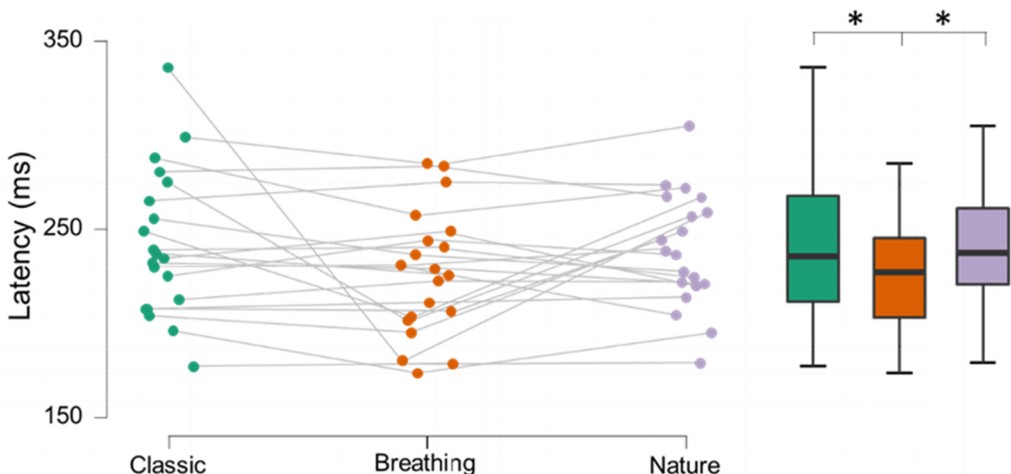

**Figure 2.** Individual and box plot data for protocol latencies in milliseconds. Breathing showed significantly lower latencies compared to classic (t19 = −1.826, *p* = 0.042) and nature protocols (t19 = −1.801, *p* = 0.044).

Breathing rate was significantly affected by protocol. A breathing rate of one would indicate perfect adherence to the breathing protocol. During the breathing protocol, participants had an average breathing rate of 1.23 (±0.38) breaths per twelve-second cycle, as compared with the average rates of the nature (2.73 ± 0.85 breaths per session) or classic (2.79 ± 0.79 breaths per session) protocol. There was a statistically significantly reduction in breathing rate with the application of the breathing protocol compared to both the other protocols (z19 = −3.883, *p* < 0.001 and z19 = −3.921, *p* < 0.001, respectively), but no difference was found between the nature and classic protocols (t19 = −0.671, *p* = 0.272).

To test the quality of the data and experience, we compared the number of remaining stimulations, from a possible 20, after artifact correction (see Section 2.4). The breathing protocol retained the highest average stimulations (*n* = 15.75 ± 3.90), followed by the classic (*n* = 15.25 ± 3.48) and nature (*n* = 13.75 ± 4.51) protocols. Only the breathing protocol showed a significant improvement over the nature protocol (z19 = −3.270, *p* < 0.001). The ease of protocol was similar for the different protocols (classic (4.48/5 ± 0.77), nature (4.25/5 ± 0.82) and breathing (4.28/5 ± 0.75)). Acceptableness was highest in breathing (4.22/5 ± 0.72), followed by nature (3.9/5 ± 0.85) and classic (3.58/5 ± 1.18) protocols. Breathing differed significantly compared to both the control protocols (z19 = −2.095, *p* = 0.018 and z19 = −2.295, *p* = 0.011, respectively). Of the 20 participants, 65% of participants chose the breathing protocol as their favorite, with 25% choosing the nature video and 10% choosing the classic protocol, the proportions of which were significant according to random chance ($\chi^2$ (2, *n* = 20) = 49.91, *p* < 0.001).

## 4. Discussion

Using LEP stimulation variants, we demonstrate that the latencies of the LEP response are shorter using a protocol with controlled breathing. We found no difference in amplitudes. Even just watching a slow, calming video of a nature scenery decreased LEP latencies, but the effect of breathing was larger. Shorter latencies might be closer to the actual minimum transduction velocity of the small nerve fibers and central processing, which would allow for a better clinical classification of nerve fiber damage or dysfunction in patients with a variety of pain conditions [4]. In clinical practice, latency is a more reliable parameter than amplitude. Due to large intra- and interindividual variability, large axon loss may occur without the amplitude decreasing below normal values. In a recent study using transspinal direct current stimulation, or tsDCS, LEP latency reflected pain pathway signaling more effectively than LEP amplitude; the amplitudes proved more placebo sensitive [26]. Furthermore, LEP latencies have been shown to be less affected by factors such as age [14].

Latencies during instructed breathing were shorter compared to both the control conditions, the classic clinical protocol and the nature video. The effect was small, but this initial finding suggests that there is the potential for further improvements in measuring more consistent LEP signals. Future studies could, for example, test phase locking the LEP stimulation to the breathing and/or the heart rate cycle. In our analysis, we found that the nature video condition also led to significantly shorter latencies than the classical protocol. Interestingly, it has been suggested previously [27] that the viewing of nature scenes following stress contributes to the balancing of autonomic function quantified as heart rate variability. This indicates that the viewing of a nature video in our study might have caused the participants' hearts rates to increase in variability, regardless of their lower breathing rates. In the instructed breathing, we found shorter latencies in response to the LEP stimulation.

We did not find an effect on the amplitude of the neural response, as has been reported in previous studies using somatosensory evoked potentials locked to cardiac cycle 10. This indicates that the interaction effects of respiration with the perception and processing of LEP stimuli might differ from what has been observed for the cardiac cycle. It could also be related to the principal differences in the measurement parameters of SEPs compared to LEPs [4]. However, we did not use a phase-locked protocol here, which might be necessary for evoking higher amplitudes [23].

There are few comparable studies. One study used ventilator-controlled breathing and found reduced amplitudes in response to LEP stimulation [22]. However, this study tried to mimic air hunger and restricted breathing instead of enhancing slow and deep breathing as we demonstrated here. Another study found a dependency of amplitude, but not latency, on the breathing phase [23]. Here, potentials were evoked through painful intradermal needle stimulation, not by LEPs, and subjects breathed normally, and did not perform guided breathing as in our design. Amplitudes differed between inhaling and exhaling—a comparison that we did not include in our design, as we did not set out to compare different phases in the respiratory cycle but the efficacy of a possible improvement of a clinical protocol based on recent findings on the interaction of respiration with evoked potentials. Our subjects were instructed to breath in a regular and slow way, which has been shown to activate the vagal nerve and the parasympathetic nervous system [28]. Such slow breathing paradigm has been used to reduce pain perception and negative effects during pain perception [29], further suggesting that an instructed slow breathing could improve the clinical use of LEPs.

The breathing protocol was the favored protocol. Many participants reported either that it was a distraction from the discomfort, or that the task itself was challenging enough to not notice the full intensity of the stimulation. The acceptableness ratings backed up the anecdote that the breathing protocol was a more comfortable experience. This did not compromise the quality of the collected data, especially compared to just directing their attention to a passive task, as in the nature protocol.

Through which mechanism the neural responses to LEP stimulation depend on the respiratory cycle phase remains to be elucidated. Notwithstanding, our results provide new evidence that clinical protocols using LEPs for the characterization of small nerve fiber dysfunction and pain signaling could be improved by adding the task of guiding patients' breathing—a simple, low-cost addition to improve the diagnostic capability of LEPs.

## 5. Conclusions

Using a simple, low-cost breathing protocol, the standard LEP protocol can be improved. By slowing participants' breathing rate and standardizing focus on a breathing task, participants showed significantly shorter LEP latencies compared to a control condition (a nature video) and the classic LEP protocol. Our simplified, low-cost protocol can be considered as either an add-on or stand-alone protocol to improve the diagnostic capability of LEPs.

**Author Contributions:** Conceptualization, A.W. and R.B.; methodology, A.W.; formal analysis, A.W.; investigation, A.W.; resources, M.T.; writing—original draft preparation, A.W. and R.B.; writing—review and editing, A.W., R.B. and M.T.; visualization, A.W.; supervision, M.T.; project administration, A.W., R.B. and M.T. All authors have read and agreed to the published version of the manuscript.

**Funding:** This research received no external funding.

**Institutional Review Board Statement:** The study received ethical approval (Dnr 2016/433-31) and was conducted in accordance with the Helsinki declaration.

**Informed Consent Statement:** Informed consent was obtained from all subjects involved in the study.

**Acknowledgments:** We would like to thank Mats Svantesson and Ilona Croy for their programming contributions.

**Conflicts of Interest:** The authors declare no conflict of interest.

## Abbreviations

| | |
|---|---|
| ECG | Electrocardiogram |
| LEPs | Laser evoked potentials |
| SEPs | Somatosensory evoked potentials |
| tsDCS | Transspinal direct current stimulation |

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
