# Peer review of "Just Breathe: Improving LEP Outcomes through Long Interval Breathing"

_ctn, doi:10.3390/ctn6020013_

Round 1

Reviewer 1 Report

In my opinion the finding of the reduced latency of LEP after breathing protocol should be further clarified; practically, what is the meaning of a shortening of LEP latency without a LEP amplitude increasing?Why the authors assert that this finding allow for a better classification of small fiber damage or dysfunction?Why there is the potential for further improvements in measuring more consistent LEP signals?Indeed, LEP amplitude and not latency is the most important parameter to detect a small fiber damage, and only few reports demonstrated an increase in latency in certain peripheral nerve diseases, such as hereditary neuropathies. 

Author Response

Dear Reviewer, 

Thank you for your feedback. I've addressed everything you commented on. Here are my responses to your highlighted review:

Introduction:

Clinical application of LEPs should be stressed in this section. Diagnostic accuracy in small fiber neuropathy should be cited (Di Stefano et al, Pain, 2018).

We thank the reviewer for this suggestion. We added in the first part of the introduction two citations about clinical applications (Truini A, et al. Pain, 2010 and Di Stefano, et al. Pain, 2017), and added the following text into: LEP “… is considered the most accurate diagnostic test for selectively assessing the nociceptive system, especially for investigating patients with neuropathic pain conditions, such as small fiber neuropathy”.

Scientific background and rationale for the investigation should be clearly reported

We agree that the introduction could expanded on the rationale. In the 3rd paragraph, we now emphasize the potential of a breathing protocol for improving LEP with the following text: “Depriving breathing in an air hunger LEP study showed an effect on LEP amplitudes (Dangers et al. 2015), providing a mechanism through which air satiation could possibly improve LEP outcomes.” We also stated the hypothesis more clearly: “In other words, following a breathing instructional video should provide an improvement within the two main outcome variables, amplitude and latency, over the classic LEP protocol. We therefore set out to compare a classical clinical protocol of LEP with a variant of this protocol where subjects were instructed to breath at a certain rate. As a control condition for potential effects of the calm video which instructed the breathing, subjects watched a nature video without breath guidance.”

Methods:

- Outcome variables should be clearly reported in this section.

Thanks for pointing this out. In section 2.4, we added the outcomes variables with the following text: “Our outcomes of interest are N2-P2 amplitude in microvolts and shortest peak N2 latency in milliseconds. Our further measures of interest are the amount of breathes taken within each twelve-second stimulation cycle, number of artifact-free cycles (of a possible 20), rating on ease of protocol and acceptableness, and which was the favored protocol”.

- Inclusion and exclusion criteria should be reported

We clarified this in the methods section 2.1: “The inclusion criteria were a self-reported full health nerve status and toleration of the lowest LEP setting. Exclusion criteria were previous nerve damage in either hand, arm or shoulder, or any neuropathy symptoms in the upper extremities (tingling, numbness, or temperature sensitivities).”

- Latency and amplitude of the lateralized N1 component are not mentioned

Sadly, we don’t see N1 components very often with our LEP set-up. In our set-up, and most others, N2 and P2 elements are the most reliable and replicable parameter, so we stuck with those.

- Was a baseline correction applied?

No, but this is a within-subjects design. We added that there was no baseline correction into the methods section 2.4.” No baseline correction was performed.”

Results:

The quality of the picture should be improved

 The editor said they would resolve the picture quality.

Data on artifacts rejection are not provided in this section

This was an important aspect to have left out, thank you for noticing. We added the following text to the manuscript: “Our script has a built-in blink detection artifact filter for any peak eye channel stimulus greater than 100μV occurring between 100 to 500ms post stimulus. Visual inspection was used to determine if a blink artifact could have influenced a recording that was outside of automatic thresholding, for example, a blink before 100ms that still effected the 100 to 500ms recording window, in which case it too was removed. Where artifacts occurred, we removed the stimulation from the total set and took the corrected average for max N2-P2 amplitude and shortest N2 latency.”

Reviewer 2 Report

In this paper the Authors tested the effect of a breath protocol on LEPs parameters. The breath protocol produced significantly shorter latencies as compared with the classic protocol. The topic is innovative but the real impact on clinical practice is still unclear. I have the following suggestion to improve the quality of the manuscript:

Introduction:

  • Clinical application of LEPs should be stressed in this section. Diagnostic accuracy in small fiber neuropathy should be cited (Di Stefano et al, Pain, 2018).
  • Scientific background and rationale for the investigation should be clearly reported

Methods:

- Outcome variables should be clearly reported in this section.

- Inclusion and exclusion criteria should be reported

- Latency and amplitude of the lateralized N1 component are not mentioned

- Was a baseline correction applied?

Results:

  • The quality of the picture should be improved

  • Data on artifacts rejection are not provided in this section

Author Response

Dear Reviewer, 

Thank you for your feedback. Here is our response to your highlighted review:

What is the meaning of a shortening of LEP latency without a LEP amplitude increasing?Why the authors assert that this finding allow for a better classification of small fiber damage or dysfunction?Why there is the potential for further improvements in measuring more consistent LEP signals?Indeed, LEP amplitude and not latency is the most important parameter to detect a small fiber damage, and only few reports demonstrated an increase in latency in certain peripheral nerve diseases, such as hereditary neuropathies.

This is an important point, which we now address in the discussion with the following text: “In clinical practice, latency is a more reliable parameter than amplitude. Due to large intra- and inter-individual variability, large axon loss may occur without the amplitude decreasing below normal values. In a recent study using transspinal direct current stimulation, or tsDCS, LEP latency better reflected pain pathway signaling than LEP amplitude; the amplitudes proving more placebo sensitive. Furthermore, LEP latencies have been shown to be less effected by factors such as age.” 

The overall idea with the breathing protocol is that of consistency, i.e. less noise should equal better signal. Latency and amplitude can both help diagnosis and understanding of nerve function with slightly different focus, and with more research into these parameters, their specific value for clinical practice might still improve. The discussion of whether amplitude or latency are more useful bring to mind what the psychologist Piaget said when asked if nature or nurture is more important: he answered “Which is more important in determining area, length or width?”. Latency and amplitude might be seen as the length and width of nerve function.

Reviewer 3 Report

Dear Authors; I find this study topic an interesting investigation on exploring the impact of three different treatments (breathing, nature, or classic) on LEP outcome. As a licensed statistician, I think, it needs "some serious extra work" to make it publishable at MDPI. Regards. P.S.

[1] Statistical:

1-1: Low sample size: Only 20 participants were used in the study. Hence, there is a need to report the statistical power of the reported tests in the study(must be minimum 80% for each). Please report these power stats in the revised text. 

1-2:  Confounding Issues: In clinical trials, it is optimum to compare similar to similar . Hence, participants in the three arms must be similar except on the treatment status.  Which process the authors used to make sure the participants are similar in the three arms?   Propensity scores ? Matching ?  *Note that without this fundamental controlling for potential confounders, the entire study design is problematic and the results are unwarranted.  Please report on this in the revised text.

1-3: Multiple Comparison:  Did the authors check the issue of "Multiple Comparison" ? It appears we have multiple comparison here.  For example with Bonferroni Correction the theshold for significant p-value is    p_corrected=p_original/3=0.05/3=0.017.  In this case, many results in the study(including the main result) are insignificant.    Please report on this in the revised text. 

Reference:

https://en.wikipedia.org/wiki/Multiple_comparisons_problem

https://en.wikipedia.org/wiki/Bonferroni_correction

[2] Writing:

2-1 Lines 145-147:  Move them to the beginning of section "3. Results".

2-2 References: Make then in MDPI format. For example, years are in bold for articles.

2-3 Abbreviations. Make a list of used abbreviations in the manuscript right before the reference section. Example:

Abbreviations

LEP: Laser-evoked potentials; ...

Author Response

Dear Reviewer,

Thank you for your feedback. Here are our responses to your highlighted review:

[1] Statistical:

1-1: Low sample size: Only 20 participants were used in the study. Hence, there is a need to report the statistical power of the reported tests in the study(must be minimum 80% for each). Please report these power stats in the revised text.

We thank the reviewer for their comments. Our power analysis that reflects the rationale behind using 20 participants, is now reported in the manuscript with the following text: “Sample size was determined by using G*Power software (version 3.1.9.7, Dusseldorf, Germany) with parameters for a matched pair t-test design using a one-tailed alpha value of .05 and a power level of .85 with an effect size of .635, which was derived from the mean reported statistically significant effect sizes of a LEP breathing restriction study from Dangers et al. (2015).”   

1-2:  Confounding Issues: In clinical trials, it is optimum to compare similar to similar . Hence, participants in the three arms must be similar except on the treatment status.  Which process the authors used to make sure the participants are similar in the three arms?   Propensity scores ? Matching ?  *Note that without this fundamental controlling for potential confounders, the entire study design is problematic and the results are unwarranted.  Please report on this in the revised text.

After rereading the methods section, we realized that the phrasing could have been misleading. We actually used a within subjects design, not a between subjects design. All participants received all three protocols. We thank the reviewer for spotting this, and corrected the phrasing with the following text: “This was a within-subjects design with a pseudo-randomized order of the three protocols: breathing, nature, or classic (see 2.3)”.

1-3: Multiple Comparison:  Did the authors check the issue of "Multiple Comparison" ? It appears we have multiple comparison here.  For example with Bonferroni Correction the theshold for significant p-value is    p_corrected=p_original/3=0.05/3=0.017.  In this case, many results in the study(including the main result) are insignificant.    Please report on this in the revised text.

The main outcome of interest is the difference between how LEP is clinically performed (“classic protocol”) versus LEP with the breathing protocol. The guided breathing was supported by a video (an expanding and contracting circle with the words “in” and “out” on the background of a calm nature scene). To make sure that any observed effect would not simply be explainable by watching the video (e.g. simple distraction from the stimulus: attention focused on something else), we added the nature video as a control condition. Because these are considered two control conditions, we do not apply a Bonferroni correction.

[2] Writing:

2-1 Lines 145-147:  Move them to the beginning of section "3. Results".

This was added by the editor, and they said it would be corrected before publication.

2-2 References: Make then in MDPI format. For example, years are in bold for articles.

We adjusted reference format to MDPI.

2-3 Abbreviations. Make a list of used abbreviations in the manuscript right before the reference section. Example:

We added a section to the manuscript with the following text:

7   Abbreviations

ECG – Electrocardiogram

LEP – Laser evoked potential

SEP – Somatosensory evoked potential

Round 2

Reviewer 1 Report

now  I've considered the new version of paper that in my opinion can be published

Reviewer 2 Report

All issues have been solved

Reviewer 3 Report

Dear Authors; most of my main concerns were addressed satisfactorily. Regards.